# Prevalence of vertical movement asymmetries at trot in Standardbred and Swedish Warmblood foals

**Ebba Zetterberg**[ID]*, **Anna Leclercq**[ID], **Emma Persson-Sjodin, Johan Lundblad**[ID], **Pia Haubro Andersen, Elin Hernlund, Marie Rhodin**[ID]

Department of Anatomy, Physiology and Biochemistry, Swedish University of Agricultural Sciences, Uppsala, Sweden

* ebba.zetterberg@slu.se

**Data Availability Statement:** All relevant data are within the paper and its Supporting Information files.

## Abstract

Many horses, just before and during their athletic career, show vertical movement asymmetries, to the same degree as clinically lame horses. It is unknown whether these asymmetries are caused by pain or have alternative explanations, such as inherent biological variation. In the latter case, movement asymmetries would be expected to be present at a very young age. This study aimed to investigate the prevalence of movement asymmetries in foals. Motion analysis, using an inertial measurement unit-based system (Equinosis), was performed on 54 foals (31 Swedish Warmbloods, 23 Standardbreds) during straight-line trot. The foals were between 4–13 weeks old and considered sound by their owners. Differences between the vertical minimum and maximum values recorded for the head ($HD_{min}$, $HD_{max}$) and pelvis ($PD_{min}$, $PD_{max}$) between left and right stance were calculated for each stride and an average was computed for each trial. Thresholds for asymmetry were defined as absolute trial mean >6 mm for $HD_{min}$ and $HD_{max}$, and >3 mm for $PD_{min}$ and $PD_{max}$. These thresholds were exceeded for one or several parameters by 83% of Standardbred foals and 45% of Swedish Warmblood foals, demonstrating surprisingly high prevalence of asymmetries in young foals, although the risk of repetitive strain injuries and cumulative risk of trauma injuries was expected to be low in this age group. Standardbred foals showed similar prevalence of asymmetries to that reported previously for yearling Standardbred trotters, so relatively higher prevalence of movement asymmetries may be expected among trotters as a breed. In general, vertical head and pelvic movement asymmetries can be anticipated among foals considered sound by their owners. A better understanding of the aetiology of asymmetries is needed for correct interpretation of objective symmetry measurements in different populations of horses.

## Introduction

Lameness evaluation is usually the first step in diagnosing most orthopaedic disorders, and is traditionally based on subjective assessment of the horse's motion pattern. Vertical movement

**Funding:** Funded by H1747250, Swedish-Norwegian Foundation of Equine Research, http://hastforskning.se/. Awarded to MR, EH and PHA. The funders had no role in study design, data collection and analysis, decision to publish or preparation of the manuscript.

**Competing interests:** The authors have declared that no competing interests exist.

asymmetry of the head and pelvis are considered sensitive measures of forelimb and hindlimb lameness, respectively [1,2]. However, subjective assessment can be challenging, as reflected by low inter-rater agreement [3–7]. The limitations in visual perception of the human eye [8] and influence of expectation bias [9], underlines the necessity for objective methods for lameness evaluation. Development of inertial measurement units (IMUs) has made it possible to record vertical movement asymmetries wirelessly in horses. Studies based on IMUs have revealed that a majority (67–73%) of riding horses in regular training, and perceived sound by their owners, show movement asymmetries [10,11] to a similar degree as horses with confirmed pain-related lameness [12,13]. Particularly high prevalence of vertical movement asymmetries (93%) has been reported in Standardbred trotter yearlings [14]. Age, use and management of these populations vary greatly, from yearlings before initiation of training to adult horses in training. The high prevalence of vertical movement asymmetries in populations of horses in regular training [10,11,15] raises the question of unrecognised lameness and a potential welfare issue. However, the clinical significance of these asymmetries in apparently sound horses remains unclear and other explanations, such as inherent biological variation, have been proposed [11,16].

Painful orthopaedic conditions affecting the limbs are a well-documented cause of vertical movement asymmetry. Common causes in young animals include developmental diseases such as osteochondrosis (OC), which is prevalent in both Warmblood and Standardbred horses [17–21]. Clinical signs of OC often appear before one year of age [22], although transient subclinical lameness correlated with OC has been observed among very young foals [23]. A study of Warmblood foals in which they were screened regularly (at intervals of 4 weeks) from 1 to 11 months of age showed that early radiographic signs of OC disappeared in a majority of these foals, indicating that existing lesions should not be considered permanent before the age of five months [24]. Other than pain, surprisingly little is known about causes of movement asymmetry or how movement asymmetry changes during a horse's lifetime. Training in itself has been suggested as a cause of alteration in horse movement and increases in movement asymmetries [25]. Another factor suggested to influence motion pattern is inherent motor laterality or sidedness [26–30]. Motor laterality would be expected to be present at an early age, like handedness in humans [31,32], but it is not known whether it influences vertical movement symmetry. With age and training, motor laterality may increase, possibly due to conventions in handling and mounting horses [28].

Movement symmetry measurements of young foals is a relatively uncharted field. To the best of our knowledge, systematic clinic-based symmetry measurements based on vertical displacement have not been performed to date in foals. Such testing may contribute to further understanding of asymmetric movement patterns in horses. This while acknowledging potential effects of training, training-related injuries, handling effects and developmental diseases, but also that movement asymmetries may to some extent be inherent. The prevalence of movement asymmetry in foals may therefore differ from that in older horses and could be suspected to be lower among young foals. The aim of the present study was therefore to describe the prevalence of vertical movement asymmetries of head and pelvis at trot in sound young foals of two common breeds. This at an age when both the risk of repetitive strain injuries and the accumulated risk of trauma injuries can be expected to be low.

## Materials and methods

### Horses

Two study populations of foals were recruited through convenience sampling based on geographical location, to allow for measurement of the young individuals in their home environment. Standardbred trotter (STB) foals were recruited from one stud farm, while Warmblood

foals were recruited through social media advertisements and direct contact with stud farms. Foals whose owners agreed for them to participate were included if they were of age 4–14 weeks, perceived as sound by the owner and had no more than mild angular limb deformities (owner-reported or seen at clinical examination). Informed consent for data collection was obtained from the owners prior to the study. Before motion analysis, all foals were clinically examined by a veterinarian, with emphasis on clinical assessment of the locomotion system. The exclusion criteria used (based on clinical examination at the time of measurements) were: subjective lameness scores of ≥2 degrees (on an ordinal 0–5 lameness scale) during straight line trot, presence of swelling/wounds or painful reactions to palpation.

## Data collection

A commercial motion analysis system (Equinosis Q with Lameness Locator, software v1.2r, St. Louis, MO, USA) was used for movement symmetry measurement. Four IMUs were attached to the skin of the foals at the following anatomical locations; the pelvis in the midline between the tubera sacrale, the poll, the withers and the dorsal aspect of the pastern of the right forelimb. Data from the withers sensor were not used in the analysis. The sensors on the pastern were attached with self-adhesive bandage wraps and those on the upper body with double adhesive tape (Tesa) and a covering of tape. For foals that resisted sensor placement on the limbs, the pastern sensors were attached with adhesive polster (Snögg) for quick attachment. For foals fitted with halters, the poll sensor was attached to the halter. For many of the foals, the halter had to be secured in place, to prevent it from slipping back on the neck and thereby displaced from the poll. Securing the placement was done with the use of a headband or double adhesive tape placed under the halter on both sides of the mane. To avoid movement irregularity caused by sensor weight or proprioceptive stimulation, an inactive IMU was also attached to the left front pastern. If correct sensor placement was not achieved during a whole trial, if a sensor was lost or displaced, the trial was excluded. Foals were encouraged to trot by following the mare, who was trotted by hand in a straight line back and forth while the foal was either loose or led close to the mare by a second handler. Foals were trotted at their preferred speed with the goal of collecting a minimum of 20 strides. Measurements were repeated until the examiner believed that a good trial (steady trot and sufficient numbers of strides in a straight line) had been achieved. In cases where it was deemed necessary, measurements were disrupted beforehand, especially in warmer weather when some foals started to show signs of tiredness. During all measurements the foals were recorded either from the side by a researcher with a hand-held camera, or by fence mounted cameras capturing simultaneously from the side, from behind and from the front. No specific training of the foals was performed prior to the measurements.

## Data analysis

Movement symmetry data were analysed with the built-in software in the (Equinosis) system, using automated stride selection when possible. The default (Lameness Locator ® 2017 software v1.2r) stride selection criteria were: each stride sequence consisting of at least six strides, stride rate between 90% and 110% of median stride rate and frequency of vertical oscillation of the head and pelvic approximately twice that of the right forelimb motion. Videos from all measurements were scrutinised to ensure that only trot sequences were included in the analysis. In trials where the automated stride selection failed, by e.g. including canter or walk strides or by missing sequences of trot (seen mainly in trials with long periods of walk, canter or bucking), manual stride selection was performed using the videos as reference. Manually added stride sequences had to be longer than five complete strides and subjectively steady state trot as

judged from the video. Vertical displacement data were generated from recorded head and pelvis sensor vertical acceleration using signal decomposition and double integration by an error-correcting algorithm [33]. The differences between the two minimum and the two maximum vertical positions of the head and the pelvis, respectively, were calculated for each stride, resulting in four asymmetry parameters: head minimum difference ($HD_{min}$), head maximum difference ($HD_{max}$), pelvis minimum difference ($PD_{min}$) and pelvis maximum difference ($PD_{max}$). The IMU system software automatically adjusts these asymmetry parameters by dividing by the expected head or pelvis vertical displacement in that stride (the second harmonic after removing non-periodic movement), and then multiplies by a constant to transform back to a size-adjusted value.

Stride by stride data were exported to Matlab (Release 2019a, The MathWorks Inc.), where outlier removal by the same method as the IMU system software standard method was performed for the head asymmetry parameters. Each stride value was compared against the trial average value of all strides, using Mahalanobis distance. Strides where the parameter value exceeded three standard deviations (SD) from the mean (for the respective parameter) were removed in an iterative process that was terminated when no additional outliers were found. Trial mean and SD for each asymmetry parameter were calculated. By convention, negative asymmetry parameter values indicate asymmetries attributable to the left forelimb or hindlimb, while positive values indicate asymmetries attributed to the right forelimb or hindlimb.

### Statistical methods

Descriptive statistics were calculated. Thresholds recommended for clinical use by the IMU system developer were used to classify the foals, based on their asymmetry parameter values, as asymmetric or symmetric. Foals with absolute trial mean values of >6 mm for $HD_{min}$ and $HD_{max}$ and >3 mm for $PD_{min}$ and $PD_{max}$ were defined as asymmetrical, and parameters were separated into positive and negative values. To illustrate and quantify inter-stride variability, and thus level of evidence, the SD value was allocated to one of three categories for each parameter mean value above thresholds: low variability (0–50% of mean), moderate variability (50–120% of mean) or high variability (over 120% of mean), as described by Kallerud *et al.* [14].

## Results

### Study population and measurements

A total of 80 foals were recruited, consisting of 34 STB foals and 46 Warmblood foals. All Warmblood foals included in the study were registered as Swedish Warmblood (SWB) foals. For reasons shown in Figs 1 and 2, 28 foals were excluded, resulting in a total of 54 foals being included in the study (23 STB foals (12 mares, 11 stallions) and 31 SWB foals (15 mares, 16 stallions)). Age in weeks at the time of the measurement for the SWB foals was (mean ± SD) 9 ± 2.5 (range 4–13 weeks) and for the STB foals 8.2 ± 1.9 (range 5–12 weeks). All STB foals were kept on one stud farm and 10 (out of 31) SWB foals were kept on one stud farm. The remaining SWB foals had eight different owners and varied housing (range 1–2 foals/owner, median 1 foal/owner).

The intention was for the foals to trot in a straight line at their preferred speed. All foals except five were allowed to trot free with the mare, while the remaining five were trotted in hand behind the mare. A mean of two trials per foal and measurement were collected, resulting in at least one successful trial for most foals. For three foals, two trials were merged to reach enough strides. Foals with trials that did not exceed 20 trot strides, or where merging of trials would not result in enough good strides, were excluded (nine foals were excluded due to too

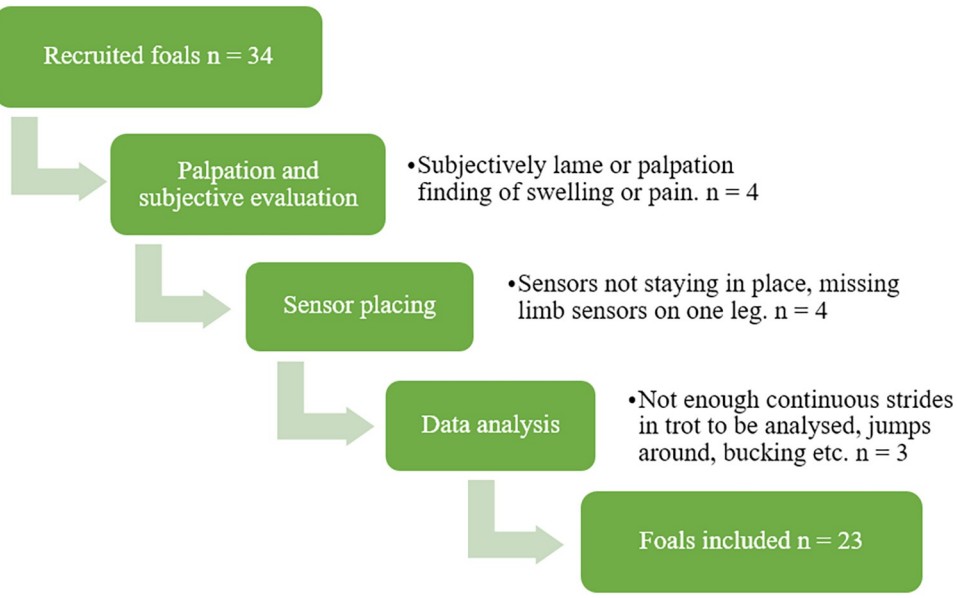

**Fig 1. Exclusion flowchart for Standardbred foals after recruitment, with number (n) of foals excluded in each section and cause of exclusion.**

few good trot strides). Number of strides collected, for trials selected for analysis, was (mean ± SD) 39 ± 14 for the STB foals and 43 ± 16 for the SWB foals. For foals with multiple trials, the first trial exceeding 20 trot strides was chosen if no other evidence indicating that the trial was unsuitable (see specifications above) was recorded.

## Descriptive statistics

The thresholds for asymmetry were exceeded for one or multiple vertical asymmetry parameters ($HD_{min}$, $HD_{max}$, $PD_{min}$ and $PD_{max}$) in 82.6% (19 of 23) of Standardbred foals and 45.2%

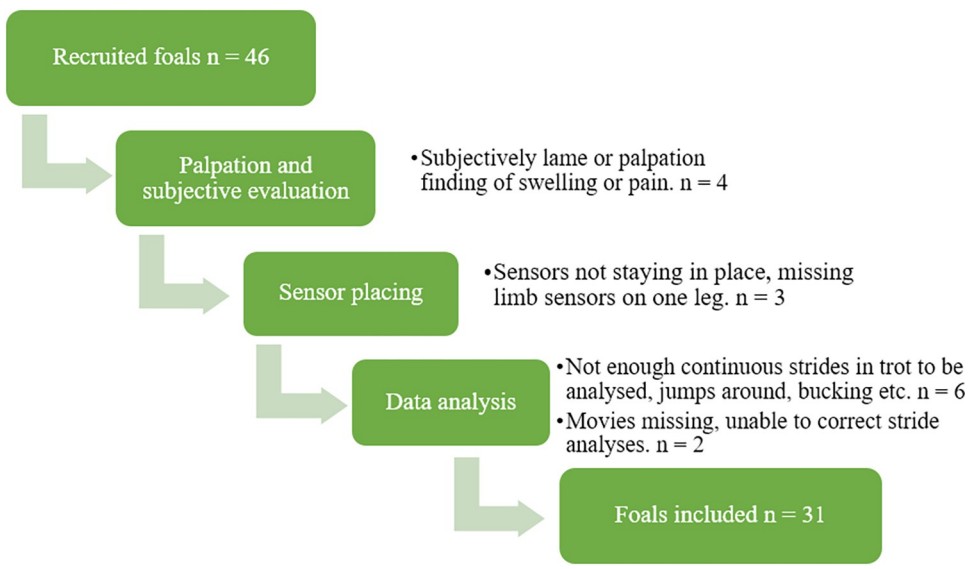

**Fig 2. Exclusion flowchart for Warmblood foals after recruitment, with number (n) of foals excluded in each section and cause of exclusion.**

**Table 1. Included trials with asymmetry parameters exceeding the threshold values.**

| Parameter, side of asymmetry | Number of trials with parameter exceeding threshold | Mean (mm) | SD (mm) | Range (mm) | Number of trials with high variability | Number of trials with moderate variability | Number of trials with low variability | Variability range in percent |
|---|---|---|---|---|---|---|---|---|
| **Swedish Warmblood foals** | | | | | | | | |
| $HD_{min}$ right | 5 | 8.76 | 2.80 | 6.13 to 12.64 | 2 | 3 | 0 | 61–206 |
| $HD_{min}$ left | 3 | 10.34 | 2.65 | -7.29 to -12.08 | 3 | 0 | 0 | 154–289 |
| $HD_{max}$ right | 1 | 8.74 | | | 1 | 0 | 0 | 185 |
| $HD_{max}$ left | 3 | 8.96 | 2.70 | -6.88 to -12.01 | 2 | 1 | 0 | 105–207 |
| $PD_{min}$ right | 3 | 4.12 | 0.73 | 3.37 to 4.83 | 1 | 2 | 0 | 99–144 |
| $PD_{min}$ left | 0 | | | | | | | |
| $PD_{max}$ right | 2 | 5.02 | 0.63 | 4.57 to 5.46 | 1 | 1 | 0 | 91–283 |
| $PD_{max}$ left | 6 | 4.57 | 1.48 | -3.12 to -7.25 | 4 | 1 | 1 | 30–150 |
| **Standardbred foals** | | | | | | | | |
| $HD_{min}$ right | 3 | 8.50 | 2.46 | 7.00 to 11.34 | 3 | 0 | 0 | 129–200 |
| $HD_{min}$ left | 6 | 13.50 | 4.78 | -8.61 to -19.38 | 3 | 3 | 0 | 79–177 |
| $HD_{max}$ right | 5 | 11.93 | 6.25 | 6.80 to 22.00 | 2 | 3 | 0 | 83–154 |
| $HD_{max}$ left | 4 | 11.08 | 2.51 | -7.33 to -12.57 | 2 | 2 | 0 | 84–390 |
| $PD_{min}$ right | 9 | 5.00 | 1.73 | 3.11 to 8.29 | 5 | 4 | 0 | 101–283 |
| $PD_{min}$ left | 1 | 8.37 | | | 1 | 0 | 0 | 141 |
| $PD_{max}$ right | 4 | 4.49 | 1.57 | 3.06 to 6.07 | 3 | 1 | 0 | 118–374 |
| $PD_{max}$ left | 3 | 5.94 | 3.55 | -3.15 to -9.94 | 1 | 2 | 0 | 68–221 |

Data from included asymmetric trials from 33 foals (SWB n = 14, STB n = 19). Left limb side asymmetry = negative values, right limb side asymmetry = positive values. Abbreviations: $HD_{min/max}$, difference in head minimum/maximum positions between right and left strides; $PD_{min/max}$, difference in pelvic minimum/maximum position between right and left stride; SD standard deviation; low inter-stride variability (SD 0–50% of mean), moderate variability (SD 50–120% of mean) and high variability (SD over 120% of mean).

(14 of 31) of Swedish Warmblood foals. To obtain more reliable results, some previous studies on vertical movement asymmetries have excluded trials with high inter-stride variability, e.g. by including only horses with mean values above thresholds when SD is less than the respective mean [11]. If such exclusion criteria had been applied in the present study (because of high inter-stride variability of asymmetry parameter values; SD >120% of trial mean), only eight STB foals and 22 SWB foals would have been retained, indicating generally high inter-stride variability of movement patterns in the study population of foals and especially within the STB population. For all parameters exceeding set thresholds, means (separated into positive and negative values) for $HD_{min}$, $HD_{max}$, $PD_{min}$ and $PD_{max}$ are presented in Table 1. Values above thresholds were equally distributed between right and left for all parameters except $PD_{min}$, for which 12 out of 13 $PD_{min}$ asymmetry values were attributed to the right hindlimb. Values above thresholds showed equal distribution of high and moderate inter-stride variability, with only one value ($PD_{max}$ for one of the SWB foals) showing low variability. Frequency distribution of foals with no, one or multiple asymmetry parameters above thresholds is presented in Table 2.

## Discussion

This is the first study based on vertical displacement measurements to investigate the prevalence of movement asymmetry in presumed sound foals aged less than 14 weeks. The results revealed that vertical movement asymmetry of the head and pelvis was common in the young

**Table 2. Frequency distribution of Swedish Warmblood (SWB) and Standardbred (STB) foals with no, one or multiple movement asymmetry parameters above thresholds.**

|  | Number of foals | Percentage of SWB/STB foals |
|---|---|---|
| **Swedish Warmblood foals** | | |
| Unilateral forelimb asymmetry | 4 | 12.9 |
| Unilateral hindlimb asymmetry | 5 | 16.1 |
| Bilateral forelimb asymmetry | 0 | |
| Bilateral hindlimb asymmetry | 0 | |
| Ipsilateral asymmetry | 1 | 3.2 |
| Contralateral asymmetry | 3 | 9.7 |
| Three limb asymmetry | 1 | 3.2 |
| No asymmetry | 17 | 54.8 |
| **Standardbred foals** | | |
| Unilateral forelimb asymmetry | 6 | 31.6 |
| Unilateral hindlimb asymmetry | 4 | 21.1 |
| Bilateral forelimb asymmetry | 0 | |
| Bilateral hindlimb asymmetry | 1 | 5.3 |
| Ipsilateral asymmetry | 2 | 10.5 |
| Contralateral asymmetry | 6 | 31.6 |
| Three limb asymmetry | 0 | |
| No asymmetry | 4 | 17.4 |

A total of 14 SWB foals and 19 STB foals were classified as asymmetrical. Bilateral forelimb asymmetry refers to cases where HDmin and HDmax are attributed to different limbs, and bilateral hindlimb asymmetry to cases where PDmin and PDmax are attributed to different limbs. Ipsilateral asymmetry refer to asymmetries of both fore- and hindlimb of the same side, while contralateral asymmetry refer to fore-and hindlimb of opposite sides. Three limb asymmetry indicates to only one limb with values below thresholds.

foals studied, with an observed prevalence of 82.6% in Standardbred foals and 45.2% in Swedish Warmblood foals when applying thresholds commonly used in clinical practice. This contradicts the theory of low prevalence among young foals.

The aim in this study was to investigate movement symmetry in sound foals, so foals with any orthopaedic clinical sign, such as subjective lameness, were excluded. However, it can be challenging to evaluate foals clinically due to their behaviour and a reaction does not necessarily imply pain or dislike, as discussed by Hunt and Baker [34]. It is also not known whether signs of radiographic OC, or other skeletal abnormalities, were present among the foals in this study, as no radiographic screening was performed. Therefore, some common disorders in the musculoskeletal system might not have been registered during clinical examination, which may have influenced the results. Clinical signs of OC in foals are considered to be present mainly from the age of 7 months [35]. However, Gorissen *et al.* [23] detected subclinical lameness among young foals as early as 4 weeks of age, in relation to seemingly temporary radiographic signs of OC. The foals in that study were considered subjectively to be sound, but had a decrease in peak vertical force of the radiologically affected leg, indicating subclinical lameness [23]. Decreased peak vertical force has been shown to be correlated with changes in the movement asymmetry parameters [36] used in the present study, but it is difficult to deduce whether all the changes in peak vertical force seen by Gorissen *et al.* would result in measurable vertical movement asymmetries above set thresholds. An OC prevalence study on STB trotters in Norway found osteochondral lesions in 50.7% (184 of 363) of the horses [19], which is a considerably lower prevalence than that of vertical movement asymmetries among yearling STB trotters [14] and in our STB foals. Considering the limitations in subjective evaluation

and the potential presence of subclinical lameness, OC might explain some of the asymmetries detected in the present study. However, the reported prevalence of OC is considerably lower than the prevalence of asymmetries, so OC is most likely not the sole reason for the asymmetries observed.

The high prevalence of $HD_{min}$, $HD_{max}$, $PD_{min}$ and $PD_{max}$ values with high inter-stride variability could indicate presence of asymmetries not associated with pain. In horses with induced lameness, studies have shown a significantly lower variation between strides [37], and similarly, orthopaedic pain has been shown to be associated with lower stride variability [38]. It may be assumed that variation in motion in a sound horse does not lead to pain, and that external stimuli or other influences are likely to influence the motion pattern to a higher extent [38]. The high inter-stride variability seen in our study can alternatively be explained by foal age. Foal gait matures during growth and variability of the gait decreases with age [23]. One study presents how Warmblood foals under 21 weeks of age do not have a clear four-beat rhythm at walk, as the diagonal and lateral duration are not equal [39], but trot development was not investigated in this study. The optimal age for conducting movement symmetry measurements could be debated, e.g. older age might be preferable in relation to research on gait development. However, we chose an age at which the foals were considered old enough to be able to trot continuously for a sufficient period, but not so old as to increase the accumulated risk of trauma injuries and the risk of repetitive strain injuries.

Subjective comparison of prevalence among our foals and older horses revealed higher prevalence of asymmetries among the older Warmblood horses [11] compared to the SWB foals, potentially reflecting an increase in asymmetries in horses during training. Significant increase in asymmetries has been reported during intense training of trotters, with elevated values persisting for almost three years [25]. This reflects the supposed outcome based on the presumption of training resulting in an increased risk of repetitive strain injuries and higher age increasing the accumulated risk of traumatic injuries. In contrast, the high prevalence of movement asymmetry among our STB foals has also been observed in older trotters [14] and exceeded that seen in other breeds of horses, including riding horses [11] and Thoroughbreds [6]. The differences in prevalence between STBs and other horse breeds have not been statistically tested, but the prevalence among trotters is seemingly higher. The reason for this is not known but a genetic component is possible, since many STB trotters have the "gait-keeper mutation" in the DMRT3 gene, which facilitates speed and lateral gaits such as pace [40] and may be favourable in races, but also decreases e.g. scores for rhythm and balance of the trot [41]. The more flexible footfall patterns of these horses may be associated with a higher prevalence of movement asymmetries, but this matter needs to be investigated further.

An alternative explanation for alterations in motion is laterality [42]. Manifestation of a preferred left or right side has been seen among adult horses [26]. However, the potential correlation between laterality and vertical movement asymmetries has not yet been investigated and further research is needed. In the present study, 12 of 13 values above thresholds for $PD_{min}$ were toward the right side, with only one toward the left side. No other parameter had the same skewed distribution. The foals trotted to similar degree on both sides of the mare and the measurements were performed in both directions to reduce influence of environmental factors (e.g. fences). A preferred side might be discussed but further, more directed, research is needed to draw any conclusions about this skewed distribution.

Early detection of signs of lameness is essential for an early treatment and thereby a potentially more favourable prognosis. Small asymmetries are difficult to detect visually and their relevance can be questioned. Vertical movement asymmetries of the pelvis and the head are sensitive measurements and correlate well with weight-bearing lameness [1], which is why this method is used to aid in lameness evaluations of clinical cases. However, these asymmetries

are also present in high prevalence in populations of horses perceived by their owners to be non-lame [10,11,14]. As demonstrated in this study, a large proportion of very young foals not yet subjected to training may also show asymmetries. One important question is whether all of these foals should be assumed to be in pain, or whether some of them are displaying congenital alterations in their motion pattern or another alternative influencing factor. In an attempt to confirm pain as a cause of asymmetries among adult horses, one study treated riding horses in training showing asymmetries with meloxicam (a non-steroidal anti-inflammatory drug with indication for treatment of acute and chronic locomotive disorders) and found that the treatment did not significantly affect asymmetries. These results indicate pain non-responsive to meloxicam or a non-painful cause for the asymmetries [43]. If horses are born with vertical asymmetries of the head and pelvis that are unrelated to pain, it is difficult to decide how the older population of horses, perceived as non-lame by owners, but presenting with asymmetries should be evaluated. This in the context of objective gait analysis, with no specific suspicion of lameness. Taking the above into consideration, the individual horse might well serve as its own reference for evaluating objective movement measurements when not related to suspicion of lameness. The availability of smartphones and ongoing rapid technological advances may soon make it possible for owners to track their horses´ movements [44] and detect irregularities in the movement "pattern" for that specific horse [15]. The aim in the present study was to describe and present the prevalence and magnitude of movement asymmetries in foals and not to draw conclusions regarding underlying causes. It is uncertain whether the clinically used thresholds applied here are fully suitable for use as "lameness screening thresholds" among horses presumed to be sound and performing according to expectations. The use of thresholds can result in mislabelling and it might have been better not to set rigid cut-off values in our population of foals. However, we use thresholds in order to allow comparison of the results with findings in other studies.

## Limitations

The system we used for objective gait analysis in foals is designed for adult horses; the asymmetry values obtained are adjusted for the range of motion of an adult horse, and it is uncertain whether these adjustments can be translated to foals. The thresholds generally used are based on a repeatability perspective [33], but a cut-off value that completely agrees with veterinarians' verdicts of lameness in horses remains to be determined [45]. Additionally it would have been of interest to follow the foals for a longer period, to assess the repeatability of results. Repeated measurements could also provide information about inter-individual fluctuations over time, in relation to e.g. handling, training or possible development of pathologies.

The high proportion of excluded study subjects shows the challenges in performing objective motion analysis in young foals due to difficulties in controlling speed and gait, and thus displacement of sensors. The low number of subjects in this study is another limitation and the results might not be generalizable to larger populations of foals, or to other breeds. Use of a larger and more diverse sample, in combination with lateralisation tests (for example limb preference tests [26]), radiographic screenings and pain evaluations (with the use of e.g. diagnostic anaesthesia), could provide more information and might be able to highlight possible aetiologies.

## Conclusions

The prevalence of vertical movement asymmetries among Standardbred foals was found to be 83%, which is similar to that reported previously for yearling Standardbred trotters. The prevalence of asymmetries was lower (45%) in the group of Swedish Warmblood foals than reported

previously for adult Swedish Warmblood and Standardbred foals. Further studies examining vertical movement asymmetries and their aetiology are needed for reliable interpretation of objective symmetry measurements, particularly in horses perceived as sound.

## Supporting information

**S1 File. Data set from the data analysis.** Excel file with the data used for SWB foals.
(XLSX)

**S2 File. Data set from the data analysis.** Excel file with the data used for STB foals.
(XLSX)

## Acknowledgments

The authors would like to thank the owners of the foals for their participation.

## Author Contributions

**Conceptualization:** Emma Persson-Sjodin, Pia Haubro Andersen, Elin Hernlund, Marie Rhodin.

**Data curation:** Ebba Zetterberg, Anna Leclercq, Johan Lundblad.

**Formal analysis:** Ebba Zetterberg, Anna Leclercq, Emma Persson-Sjodin, Johan Lundblad, Marie Rhodin.

**Funding acquisition:** Pia Haubro Andersen, Elin Hernlund, Marie Rhodin.

**Investigation:** Ebba Zetterberg, Anna Leclercq, Johan Lundblad.

**Methodology:** Emma Persson-Sjodin, Elin Hernlund, Marie Rhodin.

**Project administration:** Elin Hernlund, Marie Rhodin.

**Resources:** Marie Rhodin.

**Software:** Johan Lundblad.

**Supervision:** Elin Hernlund, Marie Rhodin.

**Validation:** Ebba Zetterberg, Marie Rhodin.

**Visualization:** Ebba Zetterberg, Anna Leclercq, Marie Rhodin.

**Writing – original draft:** Ebba Zetterberg, Marie Rhodin.

**Writing – review & editing:** Ebba Zetterberg, Anna Leclercq, Emma Persson-Sjodin, Johan Lundblad, Pia Haubro Andersen, Elin Hernlund, Marie Rhodin.

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
