## [Decision Letter · Decision Letter 0]

9 Jan 2023

PONE-D-22-33503Prevalence of vertical movement asymmetries at trot in Standardbred and Swedish Warmblood foalsPLOS ONE

Dear Dr. Zetterberg,

Thank you for submitting your manuscript to PLOS ONE. After careful consideration, we feel that it has merit but does not fully meet PLOS ONE’s publication criteria as it currently stands. Therefore, we invite you to submit a revised version of the manuscript that addresses the points raised during the review process. The reviewers have identified some points that require clarification.  These suggested edits should improve the clarity of the manuscript to a wider audience.

We look forward to receiving your revised manuscript.

Kind regards,

Chris Rogers

Academic Editor

PLOS ONE

Journal Requirements:

Reviewers' comments:

Reviewer's Responses to Questions

**Comments to the Author**

1. Is the manuscript technically sound, and do the data support the conclusions?

Reviewer #1: Yes

Reviewer #2: Partly

2. Has the statistical analysis been performed appropriately and rigorously? 

Reviewer #1: N/A

Reviewer #2: No

3. Have the authors made all data underlying the findings in their manuscript fully available?

Reviewer #1: Yes

Reviewer #2: Yes

4. Is the manuscript presented in an intelligible fashion and written in standard English?

Reviewer #1: Yes

Reviewer #2: No

5. Review Comments to the Author

Reviewer #1: This study adequately describes the vertical symmetry of head and pelvis at the trot in young foals. The methods are well described and results are presented in sufficient detail. Please change the name of the IMU system to 'Equinoxes Q with Lameness Locator software (version...)'. The discussion is broad but generally acceptable.

Reviewer #2: This paper describes the measurement of gait asymmetries in young foals. This is a potentially important study because there is increasing recognition that measurable asymmetry does not necessarily equate with clinically significant pain-related lameness. However, the study has some limitations, not all of which have been discussed adequately. Ideally these foals should have been assessed on more than one occasion to determine the repeatability of the results. More use could have been made of the video recordings to assess movement and to discuss the factors that may have influenced the results. Young foals are generally not trained to trot in straight lines, with a straight neck and trunk, and at a constant speed - how might this have influenced the results? I think that the only conclusion that can be made is that there was a high prevalence of measurable asymmetry in these young foals; we have no idea whether similar findings would be made on a day-to-day basis, how they may be influenced by early training, nor how these results would change over the ensuing months. It is an over-interpretation in my opinion to suggest that this asymmetry reflects innate congenital alterations in their motion pattern (laterality). There are other factors which need to be considered.

The authors are also perpetuating the 'myth' that an 'owner sound horse' (dreadful term) is potentially a non-lame horse and fail to take into account other factors that could be considered in the differentiation of lame versus non-lame horses.

Key words are additional words for a search engine and should not be words in the title.

Line 15 Odd way to start the Abstract. I suggest that the first sentence is rewritten.

Lines 49-51 References needed

Line 53 Reference needed

Line 55 painful conditions .... are

Line 59 what is meant by indicated in this context?

Line 64 a horse's (not the)

Lines 73-5 Rephrase to make the meaning clearer e.g. Movement asymmetry may to some extent be inherent, although may be influenced by .....

Line 76 I would use a different word than hypothesise because your study design did not permit testing this hypothesis - unless you set a threshold for prevalence - or compared other studies statistically. You infer in the Discussion that you did test this hypothesis but in reality you did not do so in a robust way. It is nonetheless obvious that there was a high prevalence of asymmetry!

line 87 of age

Line 99 dorsal aspect of the pastern

Line 101 Source of the double adhesive tape?

Lines 104-5 'using tape under the halter lateral of the mane or a headband.' It is not clear what this means.

Line 108 How were the foals trotted? Were they free or led? Were they following their dam? This needs to be in M&M, not Results.

What prior handling had the foals received and was this consistent among the groups?

Had the foals received any prior training to trot in a straight line? How might this have influenced your results?

How were the foals encouraged to trot?

It is a shame that the foals were not examined on >1 occasion to determine the repeatability of the observations!

Line 113 Where were the cameras located - from the side, behind or in front?

Line 199 Table and figure legends should be comprehensive and should be able to be read independently of the text

Table 2 What is meant by ipsilateral asymmetry, contralateral asymmetry, three limb asymmetry? These terms need to be defined for clarity – not leaving a reader to guess, perhaps incorrectly.

Why was there more right asymmetry for PD min? – 12/13 – needs discussion

How do you interpret the observations related to bilateral forelimb an hindlimb asymmetry, defined as ‘bilateral forelimb asymmetry refers to cases where HDmin and HDmax are attributed to different limbs, and bilateral hindlimb asymmetry to cases where PDmin and PDmax are attributed to different limbs.’ This needs discussion.

Line 213 asymmetry ... was

Line 218 The aim ... was

Why were horses with a lameness grade of 1/5 included in the study? How many foals had grade 1/5 lameness?

Line 220 What is meant by 'a reaction' in this context?

Line 241 From the video recordings can you say what proportion of foals trotted straight?

Line 248 What breed of foals? How many? Examined under what circumstances?

Line 255 - 6 'Subjective comparison of prevalence among our foals and older horses revealed higher prevalence of asymmetries among the older SWB horses' - what study does this refer to?

It is rather confusing to have a sentence combining observations in Warmbloods & Standardbreds.

Line 259 Increased with respect to what?

Line 262 Thoroughbreds

Ref 10 = polo ponies - were these all Thoroughbreds?? - NO

32 Thoroughbreds, 19 Argentinian polo horses and 9 horses of unknown or other breeds

Line 271 ‘Manifestation of a preferred left or right side, referred to as idiosyncratic motor laterality, has been seen among adult horses’ 24. Murphy J, Sutherland A, Arkins S. Idiosyncratic motor laterality in the horse. Appl Anim Behav Sci. 2005;91: 297–310. doi:10.1016/j.applanim.2004.11.001

Please make it clearer what type of laterality you are referring to in this context. ‘Motor’ is not enough. The laterality described in Murphy et al. does not necessarily relate to movement asymmetry in the context of the current paper. There is a real danger that the terms laterality and handedness get confused, used interchangeably and applied to different circumstances, which creates muddle and confusion. It would be helpful if the current authors clearly define what they are inferring by laterality.

Line 277 Statement needs a reference

Line 281 the term 'owner-sound horses' is vague, non-scientific and inappropriate term which is sadly becoming prevalent in objective gait analysis literature and is potentially misleading.

A better description might be ‘horses perceived by their owners to be non-lame’ - which does not necessarily equate with non-lame!

Anyone who has done extensive lameness and performance work with polo ponies, riding horses and trotters will attest to the fact that a high proportion are lame, despite being in regular work. By far the most common reason for a horse not being recommended for purchase at a pre-purchase examination is because of lameness.

See also

Dyson, S., Bondi, A., Routh, J., Pollard, D. (2022) Gait abnormalities and ridden horse behaviour in a convenience sample of the United Kingdom ridden sports horse and leisure horse population. Equine Vet. Educ. 34, 84-95. doi: 10.1111/eve.13395

In this study the presence of pain-related lameness was strongly supported by Ridden Horse Pain ethogram scores. These results supported those of an earlier, smaller study.

Dyson, S., Pollard, D. (2020) Application of a Ridden Horse Pain Ethogram and its relationship with gait in a convenience sample of 60 riding horses. Animals 10, 1044; doi:10.3390/ani10061044

Line 287 This is a potentially spurious & misleading argument because meloxicam is not a particularly effective drug in improving orthopaedic pain.

See review papers that have also discussed that a negative response to phenylbutazone does not preclude orthopaedic pain and the need to treat to long enough to perform and effective trial and record before, during and after.

Moreover, your own group presented a study at ICEEP in 2022 which highlighted the limitations of meloxicam:

Comparative Exercise Physiology 22 Supplement 1 S50 Analgesic testing with meloxicam did not eliminate pain in many clinically lame horses

M. Rhodin, E. Persson-Sjodin, J. Lundblad, K. Ask, P. Haubro Andersen and E. Hernlund

Swedish University of Agricultural Sciences, Anatomy, Physiology and Biochemistry, Box 7011, 75007, Sweden; marie.rhodin@slu.se

A recent study investigated if movement asymmetries in 66 riding horses were due to pain by performing analgesic testing with meloxicam. Meloxicam did not reduce the asymmetries, but it was not clear if there was pain not responsive to meloxicam or that non-painful conditions caused the asymmetries. The aim of this study was therefore to investigate if low-grade lameness not responsive to meloxicam, can be reduced by diagnostic analgesia. Twenty four 1-2 degree (0-5 scale) lame horses were recruited to the study and treated with 0.6 mg/kg meloxicam per os for seven days. Sixteen horse were still lame after treatment and a standard lameness examination including relevant diagnostic analgesia, was performed. Movement symmetry was measured using an inertial sensor system before and after diagnostic analgesia. Asymmetry variables were calculated as trial means of the stride-by-stride difference between the two displacement minima/maxima of the head (HDmin/HDmax) and pelvis (PDmin/ PDmax). While it was not possible to obtain a positive analgesic block in

5 horses, a significant reduction in lameness was seen after diagnostic analgesia in 11 horses (Wilcoxon signed-rank test, P<0.001) with a median decrease of 7.2 mm (72%) (range 3.0-35.0 mm) for at least one of the four asymmetry variables. In conclusion, seven days of meloxicam treatment of horses with subtle lameness cannot be used to exclude the presence of pain as cause of lameness. Systemic analgesic testing of horses with orthopedic disease remains a challenge for the equine practitioner and more potent treatment options need to be evaluated.

ICEEP proceedings 2022

Line 288 ‘it is difficult to decide how older populations of “owner-sound” horses with asymmetries should be evaluated’

See if asymmetries change under different circumstances e.g., in hand, after flexion tests, on the lunge on both soft and firm surfaces and ridden.

Look at behaviour!

Assess response to diagnostic anaesthesia

Greve, L., Dyson, S. What can we learn from visual and objective assessment of nonlame and lame horses in straight lines, on the lunge and ridden? Equine Vet. Educ. (2020) 32 (9): 479-491 doi: 10.1111/eve.13016

Line 292 'track their horses' movements'

Line 294 You did not define in your M&M what populations you were comparing with, nor did you test this statistically

Line 297 I unquestionably agree that the use of thresholds is potentially misleading but their use in so-called ‘owner- sound horses’ is potentially misleading as an argument for their lack of validity. It has been demonstrated that a large proportion of horses presumed by their owners to be working comfortably were lame when evaluated by skilled clinicians and had high RHpE scores, supporting the inference that these were genuine pain-induced gait asymmetries.

Line 299 Measurable pelvic asymmetry was also seen in horses assessed by experienced clinicians and judged to be non-lame in hand, on the lunge and ridden

Greve, L. and Dyson, S. (2016) Body lean angle in sound dressage horses in-hand, on the lunge and ridden. Vet. J. 217, 52-57.

They documented measurable pelvic movement asymmetry outside published data in non-lame horses (Warmbloods) assessed by experts

Line 303 an adult horse

Line 312 Please explain what you mean by lateralisation tests and provide a reference. Please also explain what you mean by 'possible pain evaluations'.

Line 313 The study could have been improved by:

Determining repeatability of results on different occasions

Longitudinal monitoring from young foals and at intervals thereafter

6. PLOS authors have the option to publish the peer review history of their article (what does this mean?). If published, this will include your full peer review and any attached files.

Reviewer #1: No

Reviewer #2: No

---

## [Author Response · Author response to Decision Letter 0]

23 Feb 2023

Thank you for the reviewer comments. I have attached the rebuttal letter named "Response to reviewers"

---

## [Editor Report · Decision Letter 1]

23 Mar 2023

Prevalence of vertical movement asymmetries at trot in Standardbred and Swedish Warmblood foals

PONE-D-22-33503R1

Dear Dr. Zetterberg,

We’re pleased to inform you that your manuscript has been judged scientifically suitable for publication and will be formally accepted for publication once it meets all outstanding technical requirements.

Kind regards,

Chris Rogers

Academic Editor

PLOS ONE

Additional Editor Comments (optional):

Thank you for your edits to the manuscript. 
---

## [Editor Report · Acceptance letter]

30 Mar 2023

PONE-D-22-33503R1 

Prevalence of vertical movement asymmetries at trot in Standardbred and Swedish Warmblood foals 

Dear Dr. Zetterberg:

I'm pleased to inform you that your manuscript has been deemed suitable for publication in PLOS ONE. Congratulations! Your manuscript is now with our production department. 

Kind regards, 

on behalf of

Dr. Chris Rogers 

Academic Editor

PLOS ONE